# Single-Isocenter Linac-Based Radiosurgery for Brain Metastases with Coplanar Arcs: A Dosimetric and Clinical Analysis

**DOI:** 10.3390/cancers15184496

**Published:** 2023-09-10

**Authors:** Valeria Faccenda, Denis Panizza, Valerio Pisoni, Sara Trivellato, Martina Camilla Daniotti, Sofia Paola Bianchi, Elena De Ponti, Stefano Arcangeli

**Affiliations:** 1Medical Physics Department, Fondazione IRCCS San Gerardo dei Tintori, 20900 Monza, Italy; valeria.faccenda@irccs-sangerardo.it (V.F.); denis.panizza@irccs-sangerardo.it (D.P.); sara.trivellato@irccs-sangerardo.it (S.T.); martina.daniotti@unimi.it (M.C.D.); elena.deponti@irccs-sangerardo.it (E.D.P.); 2School of Medicine and Surgery, University of Milan Bicocca, 20126 Milan, Italy; s.bianchi51@campus.unimib.it; 3Radiation Oncology Department, Fondazione IRCCS San Gerardo dei Tintori, 20900 Monza, Italy; valerio.pisoni@irccs-sangerardo.it

**Keywords:** brain metastases, radiosurgery, stereotactic radiotherapy, linac, volumetric arc therapy

## Abstract

**Simple Summary:**

Stereotactic radiosurgery (SRS) and fractionated SRS (fSRS) for brain metastases (BM) have become the standard of care for patients with up to 10 BM, yielding similar survival but a lower risk of long-term neurocognitive decline than whole-brain radiotherapy (WBRT). New technologies have allowed the accurate delivery of these high doses to the target using linac-based frameless techniques, and single-isocenter multiple-target (SIMT) approaches have recently been adopted for the simultaneous treatment of all BM. However, residual setup errors combined with intrafraction head motion may result in non-negligible compromised target coverage. While many groups have evaluated the efficacy of linac-based treatments using multiple and non-coplanar arcs, to our knowledge, only a few studies have presented clinical experience with coplanar arcs. Therefore, the present study aims to report our institutional investigation about the accuracy and outcomes of single-isocenter VMAT SRS treatments for both single and multiple BM delivered using only coplanar arcs and kV-CBCT as an IGRT system.

**Abstract:**

The efficacy of linac-based SRS/fSRS treatments using the single-isocenter coplanar FFF-VMAT technique for both single and multiple BM was investigated. Seventy patients (129 BM) treated with 15–21 Gy in 1 (*n* = 59) or 27 Gy in 3 (*n* = 11) fractions were analyzed. For each fraction, plans involving the intra-fractional errors measured by post-treatment CBCT were recalculated. The relationships of BM size, distance-to-isocenter, and barycenter shift with the difference in target coverage were evaluated. Clinical outcomes were assessed using logistic regression and Kaplan-Meier analysis. The median delivery time was 3.78 min (range, 1.83–9.25). The median post-treatment 3D error was 0.5 mm (range, 0.1–2.7) and the maximum rotational error was 0.3° (range, 0.0–1.3). In single BM patients, the GTV D95% was never reduced by >5%, whereas PTV D95% reductions >1% occurred in only 11 cases (29%). In multiple BM patients, dose deficits >5% and >1% occurred in 2 GTV (2%) and 34 PTV (37%), respectively. The differences in target coverage showed a moderate-to-strong correlation only with barycenter shift. Local failure of at least one treated BM occurred in 13 (21%) patients and the 1-year and 2-year local control rates for all lesions were 94% and 90%, respectively. The implemented workflow ensured that the degradation of target and brain dose metrics in delivered treatments was negligible. Along with encouraging clinical outcomes, these findings warrant a reduction in the PTV margins at our institution.

## 1. Introduction

Over the past several years, there has been a gradual transition from whole-brain radiation therapy (WBRT) to stereotactic radiosurgery (SRS) for the treatment of patients with brain metastases (BM) [1,2]. Many randomized trials reported an improved local control (LC) and overall survival (OS) when SRS was delivered with WBRT [3,4,5] and, subsequently, SRS alone yielded similar OS but higher rates of LC and a lower risk of long-term neurocognitive decline compared to WBRT + SRS [6,7,8,9,10]. Thus, international guidelines [11] now recommend SRS as the standard treatment option for patients with a limited number (1–4) of BM [12,13] and patients with a higher number of BM (5–10) with a limited cumulative volume (<15 mL) [14,15].

Because these treatments promote ablation and necrosis of the irradiated tissue, small margins, sharp dose gradients and special equipment have long been required to achieve high conformity and minimize functional brain tissue damage. New available technologies have allowed the accurate delivery of these high doses to the target also using linac-based frameless approaches [16,17,18,19,20]. Image-guided localization systems, along with noninvasive immobilization masks and full six-degree-of-freedom (6-DOF) robotic couch, were crucial in achieving this aim and led to comparable spatial accuracy of traditional invasive fixed frames [21,22,23,24], while also improving patient comfort and the planning process.

Although literature reports geometric accuracy within 1–2 mm [16,17,22,25,26,27], residual setup errors combined with intrafraction head motion may result in non-negligible compromised target coverage, undertreating the disease and risking tumor recurrence. Moreover, single-isocenter multiple-target (SIMT) volumetric modulated arc therapy (VMAT)-SRS approaches for the simultaneous treatment of all BM in a single session have been recently adopted to further shorten the treatment time. Different studies have demonstrated their consistent dosimetry and conformity compared to multiple-isocenter modalities [28,29,30,31], but their inferior robustness to rotational uncertainties has been noted, especially for small lesions at greater distances from the plan isocenter [32,33,34,35,36,37]. Consequently, assessing the dosimetric accuracy of such treatment techniques gained even more importance.

While many groups have evaluated the efficacy and safety of single-isocenter VMAT treatments using multiple and non-coplanar arcs [38,39,40,41,42,43,44,45,46], to our knowledge, only a few studies have reported clinical experience with coplanar arcs. Two comparison works [28,47] presented slightly superior plan quality for multiple non-coplanar arcs geometries, but the small differences were not deemed clinically significant in unchallenging brain BM cases. Moreover, non-coplanar planning modalities in most cases require the entrance of a staff member into the treatment room for each couch rotation and the use of dedicated and planar image-guided (IGRT) systems for patient positioning errors to be monitored and corrected at non-zero couch angles, increasing the overall treatment time and thus the probability of intra-fraction motion [39,48,49].

Therefore, the present study aims to report the geometric and dosimetric accuracy and clinical outcomes of single-isocenter linac-based SRS treatments for both single and multiple BM delivered using only coplanar arcs and kilovoltage cone beam computed tomography (kV-CBCT) as an IGRT system at a single institution.

## 2. Materials and Methods

### 2.1. Patient Cohort and Treatment Characteristic

Seventy patients with single (*n* = 38) and multiple (*n* = 32) BM treated between March 2020 and June 2022 at our department using frameless linac-based SRS were retrospectively analyzed. Patients were immobilized with conventional thermoplastic masks integrated with a homemade mouth bite for slippage reduction (*n* = 63) or a dedicated Solstat (CIVCO Medical Solutions, Iowa, US) mask (*n* = 7). The planning target volume (PTV) was obtained by a 2 mm isotropic expansion of the gross tumor volume (GTV), defined as the contrast-enhanced region in the post-contrast T1-weighted volumetric magnetic resonance imaging (MRI) sequence fused to the treatment planning CT. Both image sets were acquired with a maximum slice thickness of 1 mm.

Treatments were planned on a VersaHD linear accelerator (Elekta AB, Stockholm, Sweden) using a single-isocenter VMAT technique with multiple 6 MV flattening-filter-free (FFF) coplanar arcs. The extension and number of arcs for each plan were tailored according to the unique characteristics of the lesions. The Monte Carlo algorithm (1 mm grid spacing and 0.5% statistical uncertainty for calculation) of the treatment planning system (TPS) Monaco (Elekta AB, Stockholm, Sweden) was used for all plans. Prescribed doses were 15–21 Gy in 1 fraction (*n* = 59) and 27 Gy in 3 fractions (*n* = 11), as per current guidelines [11,50] depending on the size, the location, and the total volume of BM. All lesions were optimized to have 99% of the PTV surrounded by at least 80% of the prescribed dose and a dose gradient up to 110–115% inside the GTV. In case of target-organs at risk (OARs) overlap, the sparing of the OAR was always prioritized over full target coverage. An illustrative example of a treatment plan is reported in Figure 1.

Patients were given anti-edematous corticosteroids starting one day before SRS. Before treatment delivery, patients were positioned using three conventional fiducial marks on the mask aligned with room lasers and a CBCT scan was acquired with a fast head and neck preset of 100 kV, 20 mAs, 200° scan, 40 s acquisition time and small FOV. Following a skull-based automatic registration with the planning CT, setup errors were corrected using a robotic 6-DOF HexaPOD treatment couchtop (Elekta AB, Stockholm, Sweden). Immediately after the irradiation, a second CBCT was acquired with the same protocol and registered to the planning CT to evaluate the patient’s head motion during treatment. After SRS, all patients were followed up with a brain MRI scan every 3 months, and as needed for disease progression.

### 2.2. Dose Recalculation and Data Analysis

The translational and rotational errors recorded from the post-treatment CBCT were used to evaluate intrafractional errors. The 3D displacement and the maximal rotational deviation were calculated for each fraction. The relationships of the post-treatment errors with the delivery time and the time between pre- and post-treatment CBCT were evaluated. Pearson’s correlation coefficient and coefficient of determination (R^2^) were used to test the correlations.

The intrafractional rotations and translations were applied to the planning CT (Fx-CT) using the MIM software, version 7.1.4 (MIM Software Inc., Cleveland, OH, USA). Fx-CT was rigidly co-registered with the planning CT to transfer the original patient RT structures. The accuracy of the registration was qualitatively analyzed by using anatomical landmarks as reference points. Redundantly, an expert radiation oncologist validated the transferred structures by comparing them with the planning MRI. The resulting 3D GTV barycenter shift was calculated by taking the difference between the barycenter of each lesion in the original and Fx-CT images. The original patient-specific treatment plan was recalculated on the corresponding Fx-CT with Monaco Monte Carlo TPS, using the same algorithm and calculation properties. For fSRS treatments, the three different recalculated plans from each fraction were summed on the simulation CT to estimate the total cumulative plan. Thus, plans involving translations and rotations (Fx-plans) allowed the simulation of doses that were actually delivered to the patients. Figure 2 illustrates the dose recalculation workflow in a schematic representation.

The dosimetric accuracy of SRS treatments was evaluated by comparing the original and Fx-plans in terms of target and brain dosimetric parameters: the minimum dose to the 95% (D95%) of the volume and the mean dose were evaluated for each single GTV and PTV, and the volume (cc) that received at least 12 Gy (V12 Gy) for SRS or 20 Gy (V20 Gy) for fSRS [51], respectively, was recorded for the whole brain. The Wilcoxon-Mann-Whitney test with ties was used to assess the statistical significance (alpha = 0.05). Moreover, the correlations between the difference in target coverage between the two plans and the BM volume, maximum dimension, distance-to-isocenter (for multiple BM only), and barycenter shift were investigated, by using Pearson’s coefficient and R^2^.

Post-treatment MRI were used to assess LC according to the Response Evaluation Criteria in Solid Tumors. Time-to-event analyses for OS, brain progression-free survival (bPFS), and in-field progression-free survival (ifPFS) were estimated using the Kaplan-Meier method calculated from the end of the treatment. A logistic regression model was performed per-lesion to evaluate various treatment and tumor characteristics possibly related to better ifPFS at univariate and multivariate analyses. The variables considered were dose prescription, time between MR exam and planning, number of arcs, BM volume and maximum dimension, BM location and laterality, and original and delivered target coverage. Intergroup differences were evaluated using Fischer’s exact test (one-sided) for categorical variables and the Wilcoxon-Mann-Whitney test for continuous variables. A *p*-value < 0.05 was used for statistically significant differences. All statistical analyses were performed with the software Stata, version 9.0 (StataCorp LLC, College Station, TX, USA).

## 3. Results

### 3.1. Patient Characteristics and Plan Statistics

The main characteristics of the patient cohort with a total of 129 BM (median age, sex, histology, number of lesions per patient, and concomitant systemic therapy), as well as those of the treatment (dose prescription, multiple (1–4) coplanar arc geometries, median delivery time, and plan dose metrics) are reported in Table 1. The median single GTV volume was 0.27 cc (range, 0.01–10.48), while the single PTV had a median volume of 1.05 cc (range, 0.12–17.05). The median values of total GTV and PTV volume for each patient were 0.86 cc (range, 0.08–10.48) and 2.62 cc (range, 0.40–17.05), respectively. The median BM maximum dimension was 10.7 mm (range, 2.9–34.1). For multiple BM cases, the median distance-to-isocenter was 4.95 cm (range, 0.89–7.52).

### 3.2. Geometric Accuracy

Over all 92 fractions, the median post-treatment 3D error was 0.5 mm (range, 0.1–2.7) and the maximum rotational error was 0.3° (range, 0.0–1.3). Translations were <1 mm and <2 mm in 87% and 97% of all fractions, respectively, while >1° and >0.5° rotations were found in only 3% and 22% of the cases, respectively. The mean intrafractional errors for all the translational and rotational directions were inferior to 0.1 mm (SD < 0.5 mm) and 0.1° (SD < 0.4°), respectively. The maximum translational and rotational deviations were seen in the vertical direction (2.2 mm) and around the z-axis (1.3°), but no systematic errors above 0.5 mm and 0.4° were observed in and around any axis. Hence, the shifts had no preferential directions. For the fSRS cases, there were no relevant differences in the position accuracy during the second and third fractions; thus, all fractions were considered as individual fractions in this analysis. Weak correlations (R^2^ < 0.02) between post-treatment displacements and delivery time (beam on + gantry or collimator rotation time between different arcs) were found, with no significant differences in the mean 3D (*p* = 0.378) and maximum rotational (*p* = 0.705) errors for treatment delivery > 5 min compared to delivery < 5 min. Even considering the time between pre- and post-treatment CBCT (median = 7 min, range = 4–18 min), similar mean values of translational (*p* = 0.493) and rotational (*p* = 0.475) errors were found for duration > 10 min compared to duration < 10 min. The resulting median BM barycenter shift between the original and Fx-plans was 0.5 mm (range, 0.1–2.7). Table 2 presents the number (percentage) of fractions for single, multiple, and all BM, classified according to the magnitude of barycenter shift per-lesion: less than 1 mm, between 1 mm and 2 mm, and above 2 mm.

### 3.3. Dosimetric Accuracy

Table 3 reports the mean relative dose differences in target coverage and brain statistics between the original and Fx-plans for single, multiple, and all BM patients. None of the dosimetric comparisons yielded statistically significant differences in any of the analyzed dosimetric parameters (*p* > 0.05). In single BM patients, the GTV D95% was never reduced by >5%, and in 79% of the lesions, a loss of coverage < 1% was observed. The PTV D95% decreased by >5% in only 3 cases (8%) and a dose reduction < 1% occurred in 27 PTV (71%). In multiple BM patients, the target statistics were slightly inferior, with 2 GTV (2%) and 9 PTV (10%), for which a dose deficit > 5%, respectively, occurred. Even in these cases, variations < 1% were observed in the majority of BM for both GTV D95% (77%) and PTV D95% (63%). The GTV and PTV mean doses were minimally affected in both populations. Larger than 5% increases in brain V12–20 Gy were observed for only one single BM patient.

No statistically significant correlations (R^2^ < 0.02) were found between the differences in target coverage and the distance from the isocenter and size of the lesion (either volume or maximum dimension). Considering only multiple BM, a loss of coverage > 1% in GTV D95% was seen, respectively, in 12 and 9 lesions located at a distance of <4.95 cm and >4.95 cm from the plan isocenter and with a maximum dimension < 9.4 mm and >9.4 mm, and in 9 and 12 lesions with a volume < 0.16 cc and >0.16 cc. For single BM, the same GTV dose reduction was seen in three and five lesions with a maximum dimension < 16.1 mm and >16.1 mm, and in four lesions for both volume < 0.95 cc and >0.95 cc. The only moderate-to-strong correlation was observed with the BM barycenter shift: R^2^ = 0.70 for GTV D95% and R^2^ = 0.73 for PTV D95% for single BM and R^2^ = 0.44 for GTV D95% and R^2^ = 0.50 for PTV D95% for multiple BM.

### 3.4. Clinical Outcomes

Seven patients died within one month from the treatment due to primary disease progression. The statistical analysis was conducted on the remaining 63 patients who had at least one month of clinical and MRI follow-up (FUP). The median FUP was 8 months (range, 1–37). Twenty-one (33%) patients were still alive at the time of analysis, with a median FUP of 17 months (range, 4–37). Local failure of at least one treated BM occurred in 13 (21%) patients, while 57% of the patients experienced brain progression (in- or extra-field). Among these, 10 received salvage WBRT and nine received further SRS. After a median time of 6.4 months from SRS re-irradiation, three patients had further extra-field disease progression and received WBRT. Salvage surgery was performed in two patients due to symptomatic recurrences.

One-year and 2-year LC rates for all lesions were 94% and 90%, respectively. Per-patient rates were 89% and 84%, respectively. The median OS was 13 months, with 1-year and 2-year rates equal to 52% and 29%. The median bPFS time, 1-year and 2-year rates were 3.9 months, 40%, and 20%, respectively. The median ifPFS time was 7.8 months, while the corresponding 1-year and 2-year rates were 80% and 72%, respectively. Considering the 116 remaining lesions, 14 (12%) local failure events were observed. Univariate analysis revealed that only BM volume (GTV ≤ 0.27 vs. > 0.27 cc; OR 4.29; CI 95% 1.13–16.30; *p* = 0.032) was a significant prognostic factor for local progression. Median volumes of controlled and recurred BM were 0.22 cc and 0.76 cc, respectively. Figure 3 illustrates pre-treatment and post-treatment MRI scans of two BM.

## 4. Discussion

The actual dose delivered to 70 patients who underwent linac-based frameless brain metastases SRS or fSRS was estimated by recalculating the original plans on roto-translated CT according to intrafractional errors recorded with post-treatment CBCT.

These translational and rotational displacements included both the residual setup error and patient head motion. Weekly quality assurance test on the 6-DOF robotic couch guaranteed a re-positioning accuracy below 0.1 mm and 0.1°, confirming that a second pre-treatment CBCT was unnecessary and the contribution of residual setup errors was minimal. In this study, post-treatment shifts (referred to as intrafractional errors) were used as a surrogate for intrafraction patient motion. A single time point did not ensure a deep knowledge of how patients remained immobilized during the treatment, but relying on the stability of the thermoplastic masks to prevent unexpected or erratic head displacements, we used it as a fair indication. Other studies [48,49,52,53,54] have provided more complete intrafraction motion data by collecting X-ray images at different frequencies during the treatment, but they have reported several opposite findings. For example, Barnes et al. [53] found that 42.4% of fractions required a positioning correction greater than 0.7 mm, 0.7° during treatment and, thus, frequent intrafraction image guidance was essential to keep CTV-PTV margins small. Conversely, Lewis et al. [52] observed minimal patient motion during treatment and concluded that intrafraction imaging could be reduced without significant changes in the patient position. Likewise, Tarnavski et al. [48] showed that only a small fraction of patients would have had considerable positioning errors without intrafractional monitoring and correction, highlighting that treatment time should be minimized as much as possible to reduce the risk of positioning deviations. Because intrafractional imaging is not technically feasible in our department, we focused on decreasing the treatment duration using the single-isocenter VMAT technique, FFF beams, and coplanar geometries. In this way, most of our treatments lasted less than 5 min, which is a shorter treatment time than the majority of intrafraction patient motion studies.

The displacements recorded in the post-treatment CBCT were small, with mean intrafraction motion values < 0.1 mm and <0.1° for all translational and rotational directions. These findings are comparable to those reported in other studies investigating intrafraction patient motion through the analysis of post-treatment imaging [18,22,23,24,27,39,54], even if the use of different immobilization systems and different measuring and statistical methods might justify some heterogeneity. Babic et al. [23] evaluated intrafraction head motion for different invasive and non-invasive immobilization systems and among the non-invasive ones, they found mean 3D error values ranging from 0.45 ± 0.33 mm to 0.76 ± 0.51 mm. In the current study, only 10% of the patients were immobilized using dedicated Solstat masks, since the operators reported more difficulties in managing and handling them, and no improvements were seen after their implementation compared to the standard thermoplastic masks previously in use. Due to the unequal patient population with the two systems, no comparison between them was performed. Interestingly, our translational and rotational deviations did not correlate with the treatment time, considering both the delivery time and the time between pre- and post-CBCT scans. This may be explained by the aforementioned short duration of our treatments, although Mangesius et al. [49] observed a steep increase in 3D translational errors from 0.21 mm (SD = 0.26 mm) to 0.51 (SD = 0.35 mm) in the first 6 min of treatment.

As a consequence of these intrafractional displacements, a loss of target coverage > 1% occurred in 22% of lesions, but only in 1% of them, a dose reduction > 5% was observed. This 1% refers to 2 BM (out of 7) of a patient who had difficulties in maintaining a stable position on the couch due to a rapid worsening of its clinical status. The analysis of PTV dose variation, indicating what would have happened to the GTV with 0 mm margins, showed that 9% of them had a variation of D95% > 5%. However, neither the GTV nor PTV had a relevant dose reduction when the mean dose was considered. Overall, the dosimetric effects of the residual setup and patient head motion were negligible in our SRS treatments, proving that the safety PTV margins were effective in ensuring an adequate dose coverage of most lesions. In a simulation study, Guckenberger et al. [39] observed a target coverage decrease by >5% in 14% of cases without safety margins, but they found that a 1 mm margin was sufficient to keep the dose to the GTV within 5% in all patients. Minniti et al. [44] similarly found that a GTV-to-PTV margin of 1 mm ensured an adequate coverage of all treated lesions, while 9% of the lesions had a significantly compromised dose coverage when no additional margins were simulated. Based on the analysis herein presented and the well-known increased risk of brain necrosis using large PTV margins [55,56,57,58], we decided to reduce the PTV margins from 2 to 1 mm.

It should be noted that the results from the multiple BM population were not substantially inferior to those from the single BM population, as it was expected from the current literature on SIMT proving the great sensitivity to rotational uncertainties of this technique [32,37,40,41,42,43,44,59]. Sagawa et al. [59], assessing the dosimetric effects on HyperArc plans for both single and multiple metastases, showed that the minimum target coverage, the conformality, and the brain V10 Gy to V16 Gy significantly decreased with increasing rotational setup errors in multiple BM cases, while they remained comparable with the original plans for single BM cases. We hypothesized that our findings are related to the little rotations recorded, since most of them were smaller than 0.5°. Two retrospective studies [32,37] determined the dosimetric effects of rotational errors in SIMT treatments simulating different rotational angles about all axes and found that relevant target underdosage occurred when the rotational error increased to 2–3°, while at 0.5–1°, the D95% and V95% coverage rates values were ≥95% in the large majority of targets. Moreover, Selvan et al. [37] highlighted that the targets in which V95% was reduced by more than 5% were in the volume range of 0.05–0.07 cc and were at a radial distance of 6.2–7.2 cm. Indeed, target volume and distance to isocenter are reported also by many other groups [40,44,54] to be statistically significant predictors of target coverage. Agazaryan et al. [54] presenting the UCLA experience with SIMT suggested the use of variable margins, with larger margins for targets located > 6 cm away from the isocenter to minimize the risk of compromising the coverage. However, two clinical studies [43,60] assessing the correlation between distance from isocenter and local failure rejected the former hypothesis, confirming that the SIMT technique could be safely and effectively delivered to multiple locally dispersed BM. In our population of multiple BM patients, we did not find any correlation between the loss of target coverage and distance from isocenter or volume of the lesion. A possible explanation for this result is that the 2 mm margins are larger than those used in the clinical practice of many other centres, and the dose gradients in our coplanar VMAT SRS plans may be shallower than those achievable with other dedicated machines or non-coplanar arcs. With a larger dose distribution, as well as with predominant translational errors, all lesions had similar probability to be missed due to intrafractional errors irrespective of the volume and the distance from the isocenter. Thus, there was no reason to use different strategies for the two patient populations or for the most distant lesions.

Several studies [28,29,38,41,47,61,62] assessed that non-coplanar multiple-arc geometries (up to six arcs with three to four couch angles) are superior in terms of dose fall-off, conformality, and healthy brain volume receiving dose. However, the use of fewer couch angles or arcs is encouraged for uncomplicated BM cases to reduce treatment time, enhance patient tolerance, and limit intrafraction motion. As in the current study, Lau et al. [29] used mostly single or double coplanar arcs and found encouraging initial clinical outcomes (LC rates at 6 and 12 months equal to 91.7% and 81.5%, respectively). Our clinical results revealed that at 12 and 24 months, 94% and 90% of the lesions, respectively, were successfully controlled and that, overall, only 12% of them recurred. These findings are similar to those reported by Palmer et al. [43] and Minniti et al. [44] using 5 to 10 non-coplanar arcs. Moreover, Aiyama et al. [63] investigated the clinical significance of conformity index (CI) and gradient index (GI) in 925 patients and showed that lower CI was associated with lower local progression rates and GI did not impact complications rates, thus rejecting the hypothesis that higher CI and lower GI result in better post-SRS outcomes.

Based on these findings, our single-isocenter coplanar FFF-VMAT treatment technique was clinically acceptable and the entire implemented workflow was effective in ensuring no significant degradations of dose metrics due to residual setup and head motion errors for both single and multiple BM patients. So far, we decided to reduce the PTV margins from 2 to 1 mm and to further fasten the treatment delivery by optimizing all plans with only one single arc with a 90° collimator. Further research and prospective studies are warranted to validate these results and explore additional factors influencing treatment efficacy.

## 5. Conclusions

The dosimetric effects of residual setup and patient head motion errors have been investigated in single-isocenter VMAT SRS treatments for both single and multiple (up to 7) BM patients using only FFF-coplanar arcs and kV-CBCT as an IGRT system. The implemented workflow ensured that the degradation of target and brain dose metrics in delivered treatments was negligible, warranting a reduction in the PTV margins and in the number of arcs. Linac-based SRS treatments with this modality were feasible and resulted in encouraging clinical outcomes, comparable to other treatment approaches involving multiple non-coplanar arcs.

## Figures and Tables

**Figure 1 cancers-15-04496-f001:**
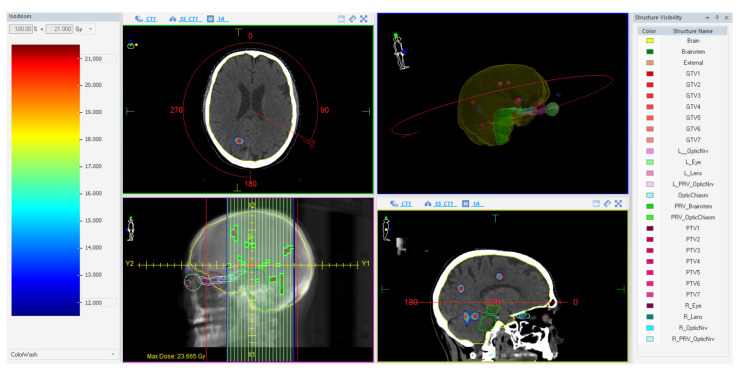
Illustrative example of a treatment plan using one almost-full single arc with a 90° collimator.

**Figure 2 cancers-15-04496-f002:**
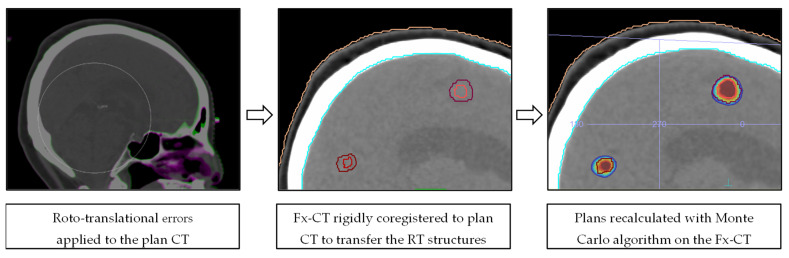
Schematic representation of the dose recalculation workflow.

**Figure 3 cancers-15-04496-f003:**
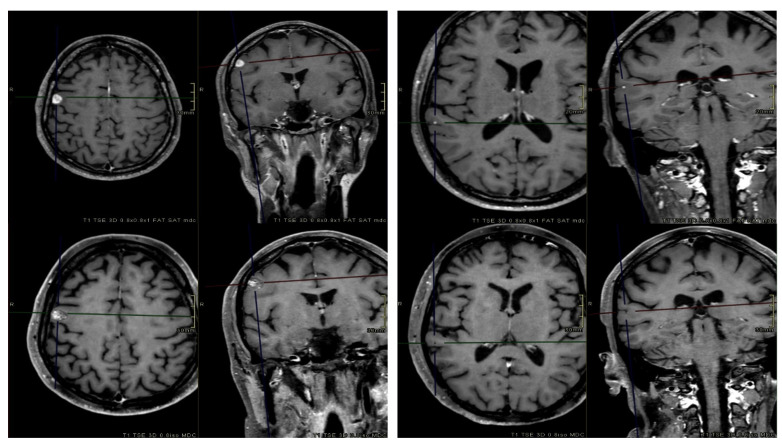
Axial and coronal contrast-enhanced T1-weighted pre-treatment (**top**) and post-treatment (**bottom**) MRI sequences of two BM. At the first MRI FUP (3 months after radiotherapy), one BM (**right**) was in progression, while the other one (**left**) completely regressed.

**Table 1 cancers-15-04496-t001:** Patient and treatment characteristics of the entire patient cohort.

Parameter		No.
Number of patients		70
Median age		66
Sex	F/M	33/37
Histology	Lung	31 (44.4%)
	Melanoma	15 (21.4%)
	Breast	12 (17.1%)
	Other	12 (17.1%)
No. of lesions per patient	1	38 (54.3%)
	2	19 (27.1%)
	3–4	10 (14.3%)
	5–7	3 (4.3%)
Concomitant systemic therapy	Chemotherapy	10 (14.3%)
	Immunotherapy	6 (8.6%)
	Molecular targeted agents	10 (14.3%)
	Hormonal therapy	2 (2.9%)
Dose prescription	21 Gy/1 fraction	46 (65.7%)
	15–18 Gy/1 fraction	13 (18.6%)
	27 Gy/3 fractions	11 (15.7%)
No. of arcs per patient (collimator angle)	1 (90°)	20 (28.6%)
	2 (0°, 90° or 45°, 315°)	38 (54.3%)
	3 (45°, 315°, 90°)	11 (15.7%)
	4 (0°, 45°, 315°, 90°)	1 (1.4%)
Median MU [range]		2644.4 [1047.0–5734.3]
Median delivery time [range]		3.78 min [1.83–9.25]
Median dose metrics [range]	GTV D95%	98.6% [73.2–105.7]
	GTV Dmean	102.8% [75.0–110.6]
	PTV D95%	87.1% [59.7–97.5]
	PTV Dmean	95.9% [70.8–102.7]
	Brain V12 Gy (1 fraction)	7.95 cc [2.23–28.46]
	Brain V20 Gy (3 fractions)	15.43 cc [8.32–23.47]

**Table 2 cancers-15-04496-t002:** Number (percentage) of fractions in which at least one BM barycenter shifted less than 1 mm, between 1 mm and 2 mm, and above 2 mm from the planned position for single, multiple, and all BM cohort.

BM Barycenter Shift	Single BM	Multiple BM	All BM
X < 1 mm	35 (73%)	111 (89%)	146 (84%)
1 mm < X < 2 mm	11 (23%)	13 (10%)	24 (14%)
X > 2 mm	2 (4%)	1 (1%)	3 (2%)

**Table 3 cancers-15-04496-t003:** Mean and range over all 129 BM and 70 patients of the percentage differences in target and brain dosimetric parameters between the recalculated and the original dose distribution. The corresponding *p*-values are also reported.

		Single BM		Multiple BM		All BM	
	Metrics	Mean [Range]	*p*-Value	Mean [Range]	*p*-Value	Mean [Range]	*p*-Value
GTV	D95%	−0.6% [−4.1–0.6]	0.666	−0.8% [−14.8–2.3]	0.639	−0.7% [−14.8–2.3]	0.556
	Dmean	0.0% [−1.1–0.6]	0.967	−0.2% [−7.3–0.8]	0.943	−0.1% [−7.3–0.8]	0.952
PTV	D95%	−1.0% [−8.7–4.0]	0.529	−1.3% [−18.8–7.6]	0.546	−1.2% [−18.8–7.6]	0.462
	Dmean	0.0% [−1.4–1.8]	0.950	−0.1% [−5.5–0.5]	0.944	−0.1% [−5.5–1.8]	0.989
Brain	V12–20 Gy	+0.4% [−2.0–7.7]	0.958	+0.4% [−0.6–3.6]	0.898	0.4% [−2.0–7.7]	0.935

## Data Availability

Research data are stored in an institutional repository and can be shared upon reasonable request to the corresponding author.

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
