# Peer review of "Single-Isocenter Linac-Based Radiosurgery for Brain Metastases with Coplanar Arcs: A Dosimetric and Clinical Analysis"

_cancers, 2023, doi:10.3390/cancers15184496_

Round 1

Reviewer 1 Report

NA

Reviewer 2 Report

The authors reported an institutional experience with single-isocentric Linac based SRS treatments for single and multiple Brain metastasis using coplanar arcs and kV cone beam computed tomography as IGRT system. The authors reported geometric and diametric accuracy and clinical outcomes observed with the adopted methodology. Encouraging clinical outcomes are reported. The reported study holds a high clinical impact as there is limited literature evidence for clinical outcomes with co-planar Arcs. The authors reported a very short, less than 5min treatment duration using single-isocenter VMAT technique, FFF beams, and coplanar geometries. Limited and unequal subject number is the limitation of this study. MRI scans were performed in this study but none of the MRI images demonstrating disease progression or lesion regression has been included in the manuscript. The size of figure 2 should be reduced. The authors have mentioned a good OS in their study so they should include some pre & post treatment MRI images. Overall, the manuscript is nicely written, methodology is well explained, and results are compiled with appropriate statistics. The manuscript can be accepted after minor revisions.

Reviewer 3 Report

Manuscript Number: cancers-2526453

Title: Single-isocenter Linac-based Radiosurgery for Brain Metastases with 2 Coplanar Arcs: a Dosimetric and Clinical Analysis

This study aims to report the geometric and dosimetric accuracy and clinical outcomes of single-isocenter linac-based SRS treatments for both single and multiple brain metastases (BM) delivered using only coplanar arcs in the single institution.

This is a reasonably written and well-structured manuscript but novelty is lacking. I recommend major revision of this manuscript because of major and minor drawbacks.

Major Drawbacks:

This study is not novel (see Lau et al [29]).

In this study, only coplanar arcs were used for single-isocenter linac-based SRS treatments for both single and multiple BM whereas in most publications (for example, [28, 29, 38, 41, 47]) non-coplanar arcs were used to create a steeper dose gradient and in such way better spare healthy brain tissue and other critical structures. The authors use as arguments for coplanar treatments a faster irradiation time and good clinical results without adding dosimetric parameters, such as conformity index, gradient index, to compare their results with non-coplanar publications. This referee is not convinced whether this manuscript will lead to any improvement in the scientific field. Make it more clear what is the novelty of this work and better explain the motivation.

Minor Drawbacks:

Abstract

“The median post-treatment 3D and maximum rotational errors were 0.5 mm [0.1–2.7] and 0.3° [0.0–1.3], respectively.”

The text is somewhat confusing. The values in the brackets show the range, do not they?

Introduction

Materials and Methods

Page 4, line 146: “The dosimetric accuracy of SRS treatments was evaluated by comparing the original and Fx-plans in terms of target and brain dosimetric parameters: the minimum dose to

the 95% (D95%) of the volume and the mean dose were evaluated for each single GTV and

PTV, and the volume (cc) that received at least 12 Gy (V12Gy) for SRS or 20 Gy (V20Gy) 149

for fSRS [51], respectively, was recorded for the whole brain.

Please explain why the whole brain in the current work was used for V12Gy instead of common

the whole brain- GTV.

Please explain which parameter was used to evaluate the dose fall off in the plans. If you are planning to use Paddick gradient index, please add the following reference:

Paddick I., Lippitz B. A simple dose gradient measurement tool to complement the conformity index. J. Neurosurg. 2006;105:194–201. doi: 10.3171/sup.2006.105.7.194.

This point is discussed in the Discussion Section.

Page 4, line 164: “A p-value <0.05 was used for statistically significant differences.”

Please add whether p-value one-sided or two-sided was used.

Results

Table 1: No. of arcs per patient (collimator angle)          1 (90°) 20 (28.6%)

2 (0°, 90° or 45°, 315°) 38 (54.3%)

3 (45°, 315°, 90°) 11 (15.7%)

4 (0°, 45°. 315°. 90°) 1 (1.4%)

This table means that no collimator angle optimalisation was used to improve sparing of the healthy brain tissue, Please reply.

Table 1: PTV D95% 87.1% [59.7 – 97.5]

This means that also D95% of 59.7% was clinically accepted for the PTV. For stereotactic treatments, PTV D95% of 98 or 99% usually used. Please explain this difference.

Page 6, line 183: “The median post-treatment 3D and maximum rotational errors over all 92 fractions were 0.5 mm [0.1 – 2.7] and 0.3° [0.0 – 1.3], respectively.”

See the comment above.

Table 3: PTV D95% -1.0 % [-8.7 – 4.0] 0.529   -1.3 % [-18.8 –7.6] 0.546   1.2 % [-18.8 –7.6]  0.462

Here it is again that a large reduction of the PTV coverage is clinically accepted.

Discussion

Pag 10, line 349: “Several studies [28, 29, 38, 41, 47] assessed that non-coplanar multiple-arc geometries (up to 6 arcs with 3 to 4 couch angles) are superior in terms of gradient index (GI), CI, and healthy brain volume receiving dose.”

There are more publications about using of non-coplanar arcs or non-coplanar IMRT beams (see the following references).

Gevaert T., Steenbeke F., Pellegri L., Engels B., Christian N., Hoornaert M.T., Verellen D., Mitine C., De Ridder M. Evaluation of a dedicated brain metastases treatment planning optimization for radiosurgery: A new treatment paradigm? Radiat. Oncol. 2016;11:1–7. doi: 10.1186/s13014-016-0593-y.

Petoukhova A, Snijder R, Wiggenraad R, de Boer-de Wit L, Mudde-van der Wouden I, Florijn M, Zindler J. Quality of Automated Stereotactic Radiosurgery Plans in Patients with 4 to 10 Brain Metastases. Cancers (Basel). 2021;13(14):3458. doi: 10.3390/cancers13143458.

English of this manuscript is good.

Round 2

Reviewer 3 Report

Manuscript Number: cancers-2526453_R1 

Title: Single-isocenter Linac-based Radiosurgery for Brain Metastases with 2 Coplanar Arcs: a Dosimetric and Clinical Analysis

The authors constructively answered my comments and comments of reviewer #2 and improved the manuscript. Therefore, I recommend this manuscript for publication.

 Accept